# Food Behaviour and Metabolic Characteristics of Children and Adolescents with Type 1 Diabetes: Relationship to Glycaemic Control

**DOI:** 10.3390/foods13040578

**Published:** 2024-02-14

**Authors:** Eulalia Catamo, Gianluca Tornese, Klemen Dovc, Davide Tinti, Raffaella Di Tonno, Vittoria Cauvin, Egidio Barbi, Roberto Franceschi, Riccardo Bonfanti, Ivana Rabbone, Tadej Battelino, Antonietta Robino

**Affiliations:** 1Institute for Maternal and Child Health—IRCCS ‘‘Burlo Garofolo”, 34137 Trieste, Italy; eulalia.catamo@burlo.trieste.it (E.C.); gianluca.tornese@burlo.trieste.it (G.T.); egidio.barbi@burlo.trieste.it (E.B.); 2Department of Medical Sciences, University of Trieste, 34149 Trieste, Italy; 3Faculty of Medicine, University of Ljubljana, 1000 Ljubljana, Slovenia; klemen.dovc@mf.uni-lj.si (K.D.); tadej.battelino@mf.uni-lj.si (T.B.); 4Department of Endocrinology, Diabetes and Metabolism, University Children’s Hospital, University Medical Centre Ljubljana, 1000 Ljubljana, Slovenia; 5Center for Pediatric Diabetology, A.O.U. Città della Salute e della Scienza, 10126 Torino, Italy; davide.tinti@unito.it; 6Diabetes Research Institute, Department of Pediatrics, IRCCS San Raffaele Hospital, 20132 Milano, Italy; ditonno.raffaella@hsr.it (R.D.T.); bonfanti.riccardo@hsr.it (R.B.); 7Division of Pediatrics, S. Chiara General Hospital, 38122 Trento, Italy; vittoria.cauvin@apss.tn.it (V.C.); roberto.franceschi@apss.tn.it (R.F.); 8Division of Pediatrics, Department of Health Sciences, Università del Piemonte Orientale, 13100 Novara, Italy; ivana.rabbone@uniupo.it

**Keywords:** type 1 diabetes, food preferences, food neophobia, parental feeding practices, glycaemic control, body mass index, lipids

## Abstract

Diet is an essential element of treating and managing type 1 diabetes (T1D). However, limited research has examined food behaviour in children and adolescents with T1D and their relationship to glycaemic control. This study evaluated food behaviour, metabolic characteristics and their impact on the glycaemic control of children and adolescents with T1D. Two hundred and fifty-eight participants with T1D (6–15 years, duration of diabetes >1 year) were recruited. Demographic, anthropometric and clinical data were collected. Questionnaires on food neophobia and food preferences were administered. The Child Food Questionnaire (CFQ) also assessed parental feeding practices. An analysis of food behaviour showed that food neophobia was inversely associated with the liking of vegetables, fruits, fish, sweets and carbohydrates. Moreover, by analysing parental feeding practices, an inverse association of “Pressure to eat”, “Monitoring” and “Restriction” with liking for vegetables and carbohydrates emerged. Considering glycaemic control, increased food neophobia and the parent practices “Restriction”, “Pressure to eat” and “Concern about weight” were found in participants with glycated haemoglobin (HbA1c) values >8.5%. Finally, higher body mass index (BMI) and total cholesterol values were observed in subjects with HbA1c values >8.5%. These findings contribute to a better understanding of eating behaviour, metabolic status and their complex relationship with glycaemic control.

## 1. Introduction

Type 1 diabetes (T1D) is one of childhood’s most common chronic diseases, resulting from an immune attack to the insulin-producing beta cells of the pancreas, leading to altered blood glucose levels [1]. In T1D, dietary management is essential to achieve optimal glycaemic control and prevent diabetes-related complications. Although a healthful diet (including fruits, vegetables, whole-grain foods and foods low in fat) is recommended, low adherence to these dietary guidelines was usually observed in children and adolescents with T1D [2]. Research on possible factors influencing dietary adherence in children with T1D has suggested that it may be related to a lack of specific knowledge on diabetes dietary management, parent–child mealtime behaviour and familiar influences [2]. For example, parents of children with T1D showed an altered perception of healthy eating practices identifying “healthy” and “unhealthy” foods based on their effect on glycaemic control [3].

Moreover, other aspects distinctive of the child’s development, such as transient food preferences and food neophobia or refusal, may also affect the eating behaviour of subjects with T1D and possibly their glycaemic control.

In healthy children, food preferences, food neophobia and their role in determining food choices have been extensively studied [4,5]. Food neophobia, the “fear of new foods”, is very common in children and manifests as the tendency to avoid eating new or unfamiliar foods. In particular, food neophobia in children was associated with reduced vegetables, fruit or meat consumption and preferences [6,7,8], less healthful food preferences and decreased food variety and dietary quality [9,10].

Family influences also play an essential role in developing food preferences and food neophobia, and greater child food neophobia was also associated with reduced compliance with parental recommendations in inducing eating [8,11].

Higher levels of food neophobia or fussy eating have been previously observed in patients with diseases that require nutritional recommendations as a part of their disease management, such as phenylketonuria [12] or a cows’ milk exclusion diet [13]. Food neophobia was also reported among individuals with celiac disease [14], and it may lead to avoidant/restrictive food intake disorders [15]. Moreover, in adults, food neophobia was also associated with altered health-related biomarkers and an increased risk of noncommunicable diseases such as obesity and type 2 diabetes [16]. Therefore, a better understanding of food neophobia in patients with these diseases would help in the making of dietary recommendations.

Similarly, knowledge of eating behaviour may be of great importance also for T1D and its management. In fact, in children with T1D, the impact of dietary intake and eating patterns on poor glycaemic control, a high prevalence of overweight/obesity and impaired lipid profiles have been reported. For example, lower carbohydrate and high fat intake were associated with worse glycaemic control, higher body mass index (BMI), more elevated total cholesterol and an increased risk of cardiovascular diseases [17,18,19]. Food neophobia was related to lower dietary variety, poorer diet quality and lower diabetes management adherence, although no association with glycaemic control and other metabolic measures was reported in T1D [20].

Therefore, this study aimed to investigate the impact of food behaviour (food neophobia, food preferences, parental feeding influences) on glycaemic control in young individuals with T1D. Moreover, the association of BMI and lipid status with glycaemic control was also analysed in this work.

## 2. Materials and Methods

### 2.1. Participants

This study included 258 participants (age 6–15 years) with a diagnosis of type 1 diabetes and disease duration of more than one year, recruited at Diabetes Units of IRCCS Burlo Garofolo (Trieste, Italy), Regina Margherita Children’s Hospital (Torino, Italy), Santa Chiara Hospital (Trento, Italy), IRCCS Ospedale San Raffaele (Milano, Italy) and UMC Ljubljana University Children’s Hospital (Ljubljana, Slovenia). Subjects with other types of diabetes (i.e., type 2 diabetes, monogenic diabetes, cystic fibrosis-related diabetes) were excluded.

All parents gave written informed consent for inclusion in this study.

This study was conducted according to the Declaration of Helsinki, and the protocol was approved by the following ethics committees: Comitato Etico Unico Regionale Friuli Venezia Giulia (CEUR-2018-Em-323-Burlo, Udine, Italy) and Komisija Republike Slovenije za medicinsko etiko (KME-0120-65/2019/4, Ljubljana, Slovenia).

### 2.2. Food Preferences and Food Neophobia Measures

Food liking was measured using a 42-item food questionnaire. Participants were asked to indicate their liking on a 7-point facial hedonic scale ranging from super bad (1) to super good (7) [21].

This present work analysed the following food groups: vegetables (carrots, broccoli, tomatoes, radicchio, cauliflower, spinach, green salad); fruits (e.g., bananas, pears, strawberries); carbohydrates (pasta, bread, rice, crackers); fish (tuna, salmon, fish); sweets (chocolate, cake, ice cream, biscuits, candies, whipped cream). The mean liking given by each subject to the foods belonging to a particular group was defined in each group. For all the food groups, the alpha Cronbach was ≥0.60, supporting internal reliability.

As previously reported in children and adolescents with T1D [22], food neophobia was evaluated using the Italian Children Food Neophobia Scale (ICFNS), consisting of 8 items: 4 related to neophobic and 4 related to neophilic attitudes [23].

The response to each item was based on a 5-point scale (“Very false for me,” “False for me,” “So-so,” “True for me,” “Very true for me”) with facial expressions that helped to better understand the level of agreement or disagreement for each of the 8 items.

Neophilic items were reversed, and a neophobia score was calculated by summing the ratings for each item. The score ranged from 8 to 40, with higher scores indicating a greater food neophobia level.

### 2.3. Child Feeding Questionnaire (CFQ)

All the participants’ parents completed the CFQ to assess their beliefs, attitudes and practices regarding child feeding [24,25]. Specifically, the following factors of CFQ were collected and evaluated: “Perceive Responsibility” (PR), parents’ perceptions of their responsibility for child feeding; “Concern about child weight” (CN), parents’ concerns about the child’s risk of being overweight; “Restriction” (RST), the extent to which parents attempt to restrict their child’s eating during meals; “Pressure to Eat” (PE) parents’ inclination to pressure their child to consume more food; and “Monitoring” (MN), the degree to which parents monitor their child’s fat intake. All items were measured using a 5-point Likert-type scale.

### 2.4. Personal and Metabolic Characteristics

Information such as age, gender and parental education level was obtained during a follow-up visit. Furthermore, parents’ education level was classified into lower education (elementary and lower secondary school) and higher education (including upper secondary school, university and doctorate).

Fasting lipids, glycated haemoglobin (HbA1c), disease duration, height and weight were also collected. BMI standard deviation scores (SDS-BMI) were estimated using Growth Calculator 4 software (http://www.weboriented.it/gh4/).

Glycaemic control was defined categorizing participants into three groups based on the following cut-offs of HbA1c values: HbA1c < 7.5% (<58 mmol/mol); HbA1c 7.5–8.5% (58–70 mmol/mol); and HbA1c > 8.5% (>70 mmol/mol) [26].

### 2.5. Statistical Analysis

Means and standard deviations for continuous variables and frequencies for categorical variables were employed to summarize participant characteristics.

ANOVA or chi-square tests were utilized to analyse sample characteristics according to glycaemic control.

The association of food liking (response variable) with neophobia and parental control of feeding (independent variables) was performed with linear regression analysis, adjusted for age, gender, population and SDS-BMI.

Regression analysis with the same covariates was performed to test the relationship between glycaemic control and food liking, food neophobia or parental control of feeding. Regression analysis with age, gender, population and disease duration as covariates was conducted to test the relationship between metabolic characteristics (lipids or SDS-BMI) and food behaviour and the association between these metabolic measures and glycaemic control.

Statistical analyses were performed using R software v 4.2.1 (www.r-project.org (accessed on 31 August 2022)).

## 3. Results

### 3.1. Sample Characteristics

Table 1 shows sample characteristics. Among the 258 participants, 124 (48%) were females. The mean age was 12.0 ± 2.6 years (range 6–15 years).

In our study, 71% percent (n = 183) of the participants were from Italy, while 29% (n = 75) were from Slovenia.

In our sample, 44% (n = 114) of participants had HbA1c < 7.5%; 37% (n = 96) between 7.5% and 8.5%, 19% (n = 48) >8.5%. HbA1c was associated with the male gender (chi-square_2_ = 6.9, *p*-value = 0.03), longer disease duration (F_2_ = 6.2, *p*-value = 0.002) and lower parent education (chi-square_2_ = 11.9, *p*-value = 0.0001).

### 3.2. Food Behaviour Measures and Glycaemic Control

Analysis of food-related measures showed that increased food neophobia was associated with a lower liking of the vegetable (*p*-value < 0.001), fruit (*p*-value = 0.008), fish (*p*-value < 0.001), sweet (*p*-value = 0.02) and carbohydrate (*p*-value < 0.001) food groups (Table 2). Moreover, we found an association between increased PE and lower carbohydrate liking. Greater RST and MN were also associated with lower vegetable liking (Table 2).

Regression analysis to test the relationship between food-related measures and glycaemic control showed an association with the food neophobia score (*p*-value = 0.04). In particular, as indicated in Figure 1a, individuals with T1D with HbA1c > 8.5% revealed an increased level of neophobia.

No significant association between the glycaemic control and food liking groups was found.

As regards parental control of child feeding, glycaemic control was associated with PE (*p*-value = 0.0014), RST (*p*-value = 0.0007) and CN (*p*-value = 0.010) scores. More specifically, increased RST, PE and CN emerged in individuals with T1D with HbA1c > 8.5% (Figure 1b–d).

### 3.3. Metabolic Characteristics and Glycaemic Control

An analysis of SDS-BMI and lipid values in T1D showed that, in our sample, RST (*p*-value = 0.02, beta = 0.19) and CN (*p*-value < 0.001, beta = 0.27) were associated with higher standardized BMI values, while PE was associated with lower BMI values (*p*-value = 0.0002, beta = −0.21).

No significant associations were detected between other food behaviour measures and SDS-BMI and lipid levels.

However, linear regression analysis (gender-, age-, population- and disease duration-adjusted) on the effect of metabolic characteristics on glycaemic control showed higher BMI and total cholesterol values in subjects with T1D with HbA1c > 8.5% (*p*-value = 0.04 and *p*-value = 0.0007) (Figure 2a,b).

## 4. Discussion

The findings from this study revealed a complex relationship between eating behaviour, metabolic measures and glycaemic control in T1D.

First, the results showed an influence of food neophobia and parental feeding practices on the food liking of children and adolescents with T1D.

Although food neophobia is common in children, it is usually exclusive to some foods such as vegetables, fruits, fish and meat [6,7,8]. In this present work, subjects with T1D also showed food neophobia for sweets and carbohydrates, possibly due to their restrictions on these categories of foods.

To our knowledge, the influence of food neophobia on food preferences among children with T1D has not previously been reported. In a past research work, Mameli and colleagues compared food preferences and neophobia in subjects affected by T1D and healthy controls; however, they examined different food groups, and they did not analyse the association between food neophobia and food preferences among individuals with T1D [22]. Otherwise, another work reported a relationship between food neophobia and dietary variety and diabetes management adherence in youth with T1D. However, the authors did not examine food preferences [20]. Therefore, it was impossible to compare the results of these previous works to this study’s findings.

In our work, an association of parental feeding practices, analysed through CFQ, with food preferences also emerged. In particular, an influence of “pressure to eat” (usually refers to attempts to increase consumption of healthy foods), “monitoring” and “restriction” (usually refers to overseeing and limiting access to specific foods) on vegetable and carbohydrate liking was observed. This result is in agreement with past works illustrating the role of food-related parenting practices in healthy children’s eating behaviour [27,28,29]. For example, restriction of a particular food was usually related to increased preference and intake of the restricted food [30,31]. In our cohort of children with T1D, probably linked to their strict diet plan for carbohydrate intake, parental pressure to eat was in fact associated with a decreased liking for these foods.

Then, this present work analysed food-related measures in children and adolescents with T1D according to their glycaemic control. An analysis of parental feeding practices showed increased levels of “pressure to eat”, “restriction” and “concern about child weight” by parents of individuals with T1D with HbA1c > 8.5%. CFQ has been widely used to measure parental feeding attitudes and strategies in studies of children without chronic diseases. However, to our knowledge, parental feeding practices in subjects with T1D have only been analysed in one previous study, the results of which partially agree with our findings. The authors reported a correlation of pressure to eat—but not restriction—with HbA1c values and suggested that CFQ may also represent a valid and adequate questionnaire for caregivers of children with T1D [32]. Moreover, our results support a recent work stating that parental eating behaviour may influence the nutritional and glycaemic status of children with T1D, thus suggesting that parental feeding practices may be a good target for interventions to prevent unhealthy eating in T1D [33].

Notably, we reported an association between poor glycaemic control (HbA1c > 8.5%) and an increased level of neophobia for the first time. In a previous study, although a relationship between neophobia and lower dietary variety and quality—as well as diabetes management adherence—was reported, glycaemic control as measured by HbA1c was not associated with neophobia [20].

However, previous findings suggest that food neophobia may lead to avoidant/restrictive food intake disorders and that parents of children with food neophobia or picky eating behaviour find it challenging to meet adequate dietary recommendations [34]. Thus, since subjects with T1D frequently fail to meet dietary guidelines, this eating behaviour may complicate adherence to diabetes management and thus influence their glycaemic control.

Finally, the findings in this study also reported an association between increased BMI and total cholesterol level with higher HbA1c values. Although controversial results have been published, past work on children and adults with T1D have already reported a relationship between higher BMI and increased HbA1c levels [35,36,37]. Similarly, our results confirmed previous studies reporting high lipid levels in children and adult individuals with poorly controlled type 1 diabetes [38,39,40].

Overall, these findings suggest the need for specific dietary approaches to both optimize weight management, lipid profiles and glycaemic control and to contribute to preventing or delaying chronic complications.

We also found a relationship between parental feeding practices and BMI among subjects with T1D, confirming results in the general population showing that “Pressure to eat” was associated with a lower BMI. In comparation, “Restriction” and “Concern about child weight” were related to a higher BMI [41,42,43]. Consistent with previous findings in healthy children, no association of the BMI with food neophobia emerged in this work [44,45,46].

This study has some limitations. First, the sample size is relatively small. Second, we did not consider additional factors affecting food preferences or glycaemic control, such as economic status, physical activity, child personality and quality of life. Third, we did not assess the presence of eating disorders nor did we collect food intake measures.

Despite these limitations, the strength of this study is that it comprehensively assesses food behaviour, parental feeding practices, glycaemic control and metabolic measures among children and adolescents with T1D. Our results may help better understand the relationships among these variables in a population where diet is essential to disease management. Therefore, these findings could contribute to guiding parents and health care practitioners toward possible interventions to prevent the development of disordered eating among children with T1D.

Further studies with larger samples and analysing the influence of additional factors are needed to better understand the relevance of food behaviour in the glycaemic control of the disease. Moreover, similar studies in different populations should be conducted to confirm our results.

## Figures and Tables

**Figure 1 foods-13-00578-f001:**
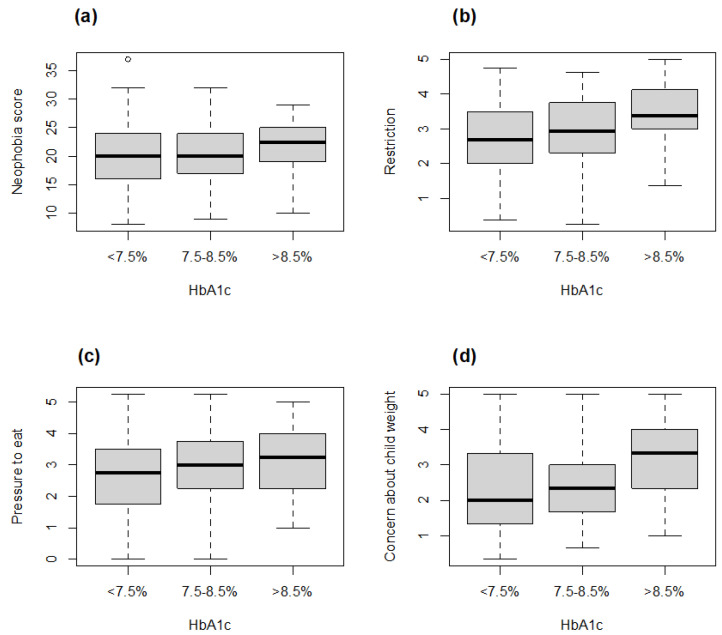
Neophobia (**a**) and parental control of child feeding ((**b**) Restriction; (**c**) Pressure to eat; (**d**) Concern about child weight) according to HbA1c.

**Figure 2 foods-13-00578-f002:**
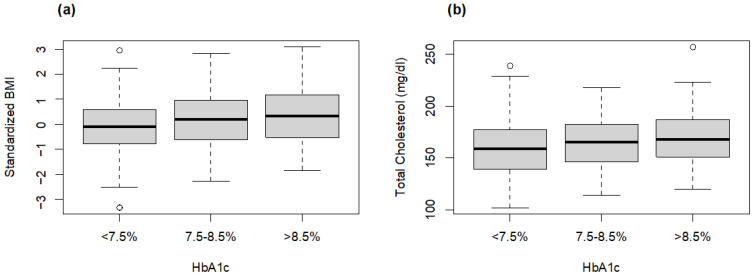
Standardized BMI (**a**) and total cholesterol (**b**) levels according to HbA1c.

**Table 1 foods-13-00578-t001:** Sample characteristics.

Characteristics	All (n = 258)	Males (n = 134)	Females (n = 124)	*p*-Value *
**Age** (years) (mean ± sd)	12.0 ± 2.6	12.4 ± 2.6	11.5 ± 2.6	0.004
**HbA1c** (%)	7.7 ± 1.0	7.8 ± 1.0	7.6 ± 1.0	0.12
**Disease duration** (years) (mean ± sd)	4.9 ± 3.2	5.2 ± 3.4	4.5 ± 3.4	0.10
**SDS-BMI** (mean ± sd)	-0.2 ± 1.1	0.1 ± 1.0	0.3 ± 1.1	0.62
**Total cholesterol** (mg/dl) (mean ± sd)	163.0 ± 28.9	158.5 ± 25.0	168.1 ± 28.0	0.005
**HDL cholesterol** (mg/dl) (mean ± sd)	63.5 ± 16.1	63.0 ± 15.6	64.1 ± 16.5	0.60
**LDL cholesterol** (mg/dl) (mean ± sd)	85.5 ± 23.2	82.1 ± 21.5	89.3 ± 24.6	0.02
**Triglycerides** (mg/dl) (mean ± sd)	71.3 ± 33.4	68.8 ± 33.9	74.1 ± 32.9	0.22

* *p*-value from *t* test.

**Table 2 foods-13-00578-t002:** Results of the association of food liking (response variable) with neophobia and parental control of feeding (independent variables).

	Neophobia	PE	RST	MN	CN	PR
*Liking group*						
Vegetables	<0.001 (−1.00)	0.10 (−0.11)	0.04 (−0.20)	0.02 (−0.20)	0.80 (−0.02)	0.60 (−0.05)
Fruits	0.008 (−0.03)	0.68 (−0.02)	0.37 (−0.08)	0.21 (−0.10)	0.26 (−0.08)	0.36 (0.06)
Fish	<0.001 (−0.07)	0.39 (−0.07)	0.20 (−0.14)	0.21 (−0.15)	0.44 (0.07)	0.74 (−0.04)
Sweets	0.02 (−0.03)	0.24 (0.06)	0.98 (−0.001)	0.55 (−0.04)	0.46 (−0.04)	0.52 (0.04)
Carbohydrates	<0.001 (−0.04)	0.03 (−0.10)	0.11 (−0.10)	0.58 (−0.03)	0.72 (0.02)	0.31 (0.06)

The values are *p*-value and beta (in brackets). In bold are the results with a *p*-value < 0.05. Models are age-, gender-, population- and standardized BMI-adjusted. PE: Pressure to eat; RST: Restriction; MN: Monitoring; CN: Concern about child weight; PR: Perceive responsibility.

## Data Availability

The data presented in this study are available on request from the corresponding author.

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
