# Peer review of "Food Behaviour and Metabolic Characteristics of Children and Adolescents with Type 1 Diabetes: Relationship to Glycaemic Control"

_foods, 2024, doi:10.3390/foods13040578_

Round 1

Reviewer 1 Report

Comments and Suggestions for Authors

Pertinence of the research:

This investigation is relevant, focusing on the food behaviour of children and adolescents with type one diabetes, whose food intake can adversely affect their health. The study focused on the young subjects (from 6 to 15 years) but also on their parents, who are responsible for preparation of their meals and ultimately food intake. Moreover, the study is based on instruments already validated and accepted by the scientific community for this type of research.

Abstract:

The abstract is well organized with a brief explanation of the purpose of the study followed by a summary of the experimental methodology followed. Then the authors presented the most relevant conclusions observed and finalize with a brief conclusion remark.

However, in the abstract there should be avoided the use of abbreviations that are not defined,. They defined some like for example T1D or CFQ, but not HbA1c. This last acronym must be identified for understanding of the results listed.

Again I alert for the need to define the abbreviation BMI on its first appearance in line 36 of abstract and line 80 if introduction when now is defined in line 87 at the end of introduction.

Introduction:

The introduction helps to frame and contextualize the work. It presents some state of the art on a number of topics which are essential to the work that was carried out, and it serves as a justification for its purpose, also highlighted at the end of introduction.

Materials and methods:

The description of the methodologies applied to obtain and treat the data are presented clearly, including all ethical issues associated with collecting and treating the data.

Results and discussion:

The results are well presented, in Tables and Figures, which are very elucidative and allow a good understanding of the data obtained.

However, I find it interesting thar the data were collected in two different countries (Italy and Slovenia), but there is absolutely no reference to that fact in the treatment of the results. Even for the sample, it is said it was collected in several places in Italy and in one other in Slovenia, but from 258 participants we do not know which percentage s from Italy and which id from Slovenia, and it would have been very interesting to understand how the country influenced results.

So, I recommend a further treatment of the data with country segmentation to add to the results already presented and this should also be included in the discussion.

Conclusions:

The conclusions part is also well formulated, presenting the most relevant findings of the work. But should be complemented with further results as recommended.

References

There is a high rate of old references, so the authors should make an effort to have more than 50% of references form the last 5 years.

Comments on the Quality of English Language

The use of English is generally OK

Author Response

Reviewer 1

This investigation is relevant, focusing on the food behaviour of children and adolescents with type one diabetes, whose food intake can adversely affect their health. The study focused on the young subjects (from 6 to 15 years) but also on their parents, who are responsible for preparation of their meals and ultimately food intake. Moreover, the study is based on instruments already validated and accepted by the scientific community for this type of research. 

We thank the Reviewer for the positive feedback on our study.

Abstract:

The abstract is well organized with a brief explanation of the purpose of the study followed by a summary of the experimental methodology followed. Then the authors presented the most relevant conclusions observed and finalize with a brief conclusion remark.

However, in the abstract there should be avoided the use of abbreviations that are not defined. They defined some like for example T1D or CFQ, but not HbA1c. This last acronym must be identified for understanding of the results listed. 

Again I alert for the need to define the abbreviation BMI on its first appearance in line 36 of abstract and line 80 if introduction when now is defined in line 87 at the end of introduction.

We corrected abbreviations as suggested.

Introduction:

The introduction helps to frame and contextualize the work. It presents some state of the art on a number of topics which are essential to the work that was carried out, and it serves as a justification for its purpose, also highlighted at the end of introduction.

Materials and methods:

The description of the methodologies applied to obtain and treat the data are presented clearly, including all ethical issues associated with collecting and treating the data.

Results and discussion:

The results are well presented, in Tables and Figures, which are very elucidative and allow a good understanding of the data obtained. 

However, I find it interesting thar the data were collected in two different countries (Italy and Slovenia), but there is absolutely no reference to that fact in the treatment of the results. Even for the sample, it is said it was collected in several places in Italy and in one other in Slovenia, but from 258 participants we do not know which percentage s from Italy and which id from Slovenia, and it would have been very interesting to understand how the country influenced results.

So, I recommend a further treatment of the data with country segmentation to add to the results already presented and this should also be included in the discussion.

We thank the reviewer for the suggestion and we apologise for the lack of information. We added the percentage of participants from Italy and Slovenia.
As regard data analysis, due the small sample size, we preferred not to split the sample by country.

However, we observed that performing statistical analysis adding country as covariate did not alter the main results and their interpretation.

In the revised version of the manuscript, we updated results according to statistical analyses performing with country as covariate.

Conclusions:

The conclusions part is also well formulated, presenting the most relevant findings of the work. But should be complemented with further results as recommended.

Discussion was slightly modified according to the updated results.

We also added in conclusions that future research in additional populations should be performed to confirm our results.

References

There is a high rate of old references, so the authors should make an effort to have more than 50% of references form the last 5 years.

We thank the reviewer for the suggestion. When it is possible, we replaced old references.

Reviewer 2 Report

Comments and Suggestions for Authors

REPORT REVIEWER

Title: Food behaviour and metabolic characteristics of children and  adolescents with type 1 diabetes: relationship with glycaemic  control.

 Dear Authors

The authors have effectively highlighted the fundamental principles of contributing to a better understanding of eating behaviour, metabolic status, and their complex relationship with glycemic control. Overall the article is well structured and the information is quite useful. Also, the authors can add this article doi: 10.3390/medicina59111954. PMID: 38004003; PMCID: PMC10673282 cite the article.

ANOVA or chi-square values should also be given in the tables where statistics are given. Performing a t-test on the age table may also show that the age was not chosen correctly in the selection of male and female samples in age matching of the sample. Figure 1, where neophobia and parental control in child nutrition are given according to HbA1c, and Figure 2, where standardized BMI and total cholesterol levels are given according to HbA1c, were found to be very positive in terms of understanding the article.

In the results, important findings should be included to make the results remarkable. The strengths and limitations of the study should be added. Information should also be given about what to do in the future. It would be useful to know what needs to be done regarding this issue in the future.

English needs to be rewritten. Some sentences are too long and need to be split into two sentences.

The study results are discussed with appropriate sources.

However, the limitation of the study is that it was planned and conducted with a small number of patients. In my opinion, it would be beneficial to increase the number of patients to increase the reliability of the study and make generalizations.

With my compliments and best regards.

Comments on the Quality of English Language

REPORT REVIEWER

Title: Food behaviour and metabolic characteristics of children and  adolescents with type 1 diabetes: relationship with glycaemic  control.

English needs to be rewritten. Some sentences are too long and need to be split into two sentences.

Author Response

Reviewer 2

The authors have effectively highlighted the fundamental principles of contributing to a better understanding of eating behaviour, metabolic status, and their complex relationship with glycemic control. Overall the article is well structured and the information is quite useful. Also, the authors can add this article doi: 10.3390/medicina59111954. PMID: 38004003; PMCID: PMC10673282 cite the article.

We thank the reviewer for the positive feedback on our study.

We also thank the reviewer for the suggested reference. Unfortunately, on close reading we could not determine exactly which part of our argument it supported. Therefore, we have not added it to the manuscript at this stage, but would welcome any specific guidance as to how it could be incorporated.

ANOVA or chi-square values should also be given in the tables where statistics are given.

We apologize for the lack of clarity. ANOVA and chi-square were used to analyse sample characteristics according to glycaemic control.

We did not show these results in tables but only in the text.
Then, in the revised version of the manuscript, we added in the text ANOVA or chi-square values.

Performing a t-test on the age table may also show that the age was not chosen correctly in the selection of male and female samples in age matching of the sample.

We are aware of this significant difference in age between males and females. In fact, age was used in all statistical analysis as covariate.

Figure 1, where neophobia and parental control in child nutrition are given according to HbA1c, and Figure 2, where standardized BMI and total cholesterol levels are given according to HbA1c, were found to be very positive in terms of understanding the article.

We thank the reviewer for the positive comment.

In the results, important findings should be included to make the results remarkable. The strengths and limitations of the study should be added. Information should also be given about what to do in the future. It would be useful to know what needs to be done regarding this issue in the future.

English needs to be rewritten. Some sentences are too long and need to be split into two sentences. 

The study results are discussed with appropriate sources. However, the limitation of the study is that it was planned and conducted with a small number of patients. In my opinion, it would be beneficial to increase the number of patients to increase the reliability of the study and make generalizations.

We agree that the small number of patients is a potential limitation. Then, as suggested, we added this limitation in the revised version of the manuscript.

Moreover, we added the following sentence on possible future research work: “Further studies with larger samples and analyzing the influence of additional factors are needed to better understand the relevance of food behavior in glycemic control of the disease. Moreover, similar studies in different populations should be conducted to confirm our results.”

The manuscript was also carefully revised by a highly qualified English speaker